# Central Neck Dissection in Papillary Thyroid Carcinoma: Benefits and Doubts in the Era of Thyroid Lobectomy

**DOI:** 10.3390/biomedicines12102177

**Published:** 2024-09-25

**Authors:** Jacopo Zocchi, Gioacchino Giugliano, Chiara Mossinelli, Cecilia Mariani, Giacomo Pietrobon, Francesco Bandi, Stefano Malpede, Enrica Grosso, Marco Federico Manzoni, Elvio De Fiori, Giovanni Mauri, Manila Rubino, Marta Tagliabue, Mohssen Ansarin

**Affiliations:** 1Department of Otorhinolaryngology and Head and Neck Surgery, European Institute of Oncology (IEO), IRCCS, 20141 Milan, Italy; jacopo.zocchi@ieo.it (J.Z.); gioacchino.giugliano@ieo.it (G.G.); chiara.mossinelli@ieo.it (C.M.); cecilia.mariani@ieo.it (C.M.); francesco.bandi@ieo.it (F.B.); enrica.grosso@ieo.it (E.G.); marta.tagliabue@ieo.it (M.T.); mohssen.ansarin@ieo.it (M.A.); 2Department of Otorhinolaryngology, Bassini Hospital, ASST Nord Milano, 20092 Cinisello Balsamo, Italy; stefano.malpede@asst-nordmilano.it; 3Onco-Endocrinology Unit, European Institute of Oncology (IEO), IRCCS, 20141 Milan, Italy; marcofederico.manzoni@ieo.it (M.F.M.); manila.rubino@ieo.it (M.R.); 4Department of Radiology, European Institute of Oncology (IEO), IRCCS, 20141 Milan, Italy; elvio.defiori@ieo.it; 5Division of Interventional Radiology, European Institute of Oncology (IEO), IRCCS, 20141 Milan, Italy; giovanni.mauri@ieo.it

**Keywords:** thyroid, neck dissection, papillary carcinoma, central neck compartment, head and neck, thyroid surgery

## Abstract

Introduction: Surgery is still the main line of treatment for papillary thyroid cancer (PTC) with a current trend for de-intensified treatment based on an excellent prognosis. The role of a routine prophylactic central neck dissection (PCND) is still debated as its impact on oncologic outcomes has never been cleared by a randomized clinical trial. In this study, we aimed to report our long-standing experience in PCND and its potential contemporary role in the treatment of PTC. Methods: A retrospective institutional review was performed on all patients who underwent operation for PTC including PCND between 1998 and 2021. The primary outcomes were the rate of central lymph node metastases (CLNMs), cancer recurrence and incidence of complications. Survivals were analyzed using the Kaplan–Meier estimator and Cox proportional hazard models. Results: A total of 657 patients were included in this study with a median follow-up of 78 months (48–114 months). Two hundred and one patients presented occult CLNMs (30.6%). The presence of a pathological node represented the unique reason for a completion thyroidectomy and I^131^ therapy in 12.5% of the population. Age lower than 55 years, microscopic or macroscopic extra-thyroid extension (ETE) and multifocality were independent factors predicting CLNMs. The rate of recurrence in the whole population was 2.7% (18 patients). Five-year and ten-year disease-free survival (DFS) was 96.5% (94.7–97.7) and 93.3% (90.3–95.5), respectively. Two patients relapsed in the central neck compartment (0.3%). Age (>55 years), pathological staging (pT) and extranodal extension (ENE) were independent factors associated with a worse DFS. The rate of temporary and permanent vocal fold palsy was 12.8% and 1.8%, respectively, and did not depend on the type of surgical procedure performed. Hypoparathyroidism was temporary in 42.2% and permanent in 11.9% of the patients. A sub-analysis upon cT1b-T2 patients treated primarily with thyroid lobectomy and ipsilateral PCND demonstrated a 2.6% rate of permanent hypoparathyroidism. Conclusions: PCND allows for a high disease-free survival and a proper selection of patients needing adjuvant treatment, in particular, those treated with a unilateral procedure. On the other hand, bilateral approach is burdened by a not-neglectable rate of permanent hypoparathyroidism.

## 1. Introduction

Treatment for papillary thyroid carcinoma (PTC) has undergone important changes in recent years. We observed a shift towards less intensive and invasive treatments owing to encouraging results reported by several institutional experiences [1,2]. Thyroid lobectomy (TL) substituted total thyroidectomy (TT) for selected (T1a–T1b–T2, N0) tumors and has been incorporated in national and international guidelines as a good clinical practice [3,4,5]. Indication for iodine radiotherapy has been reduced and now reserved for American Thyroid Association (ATA) intermediate- and high-risk group patients and selected low-risk patients [3].

In this context characterized by a tendency toward conservative and minimally invasive therapeutical strategy, the adoption of prophylactic central neck dissection (PCND) remains controversial. The rate of occult central lymph node metastasis (CLNM) in PTC is reported to be high, with some series reporting up to 80% of cases [6]. Owing to this high percentage, many authors support a routine use of PCND that guarantees a lower local recurrence rate [7], a lower rate of re-operation [8] and a proper pathological staging, and thus a correct selection of patients who need adjuvant treatment [9]. On the other hand, the impact on survival outcomes is reported to be neglectable, making the utility of PCND still debated [10,11]. At the same time, the use of PCND seems to be burdened by a higher rate of post-operative complications, in particular, permanent hypocalcemia [12].

At the European Oncologic Institute (IEO) IRCCS, we have adopted PCND since 2001 with the purpose of maximizing the locoregional control of disease and reaching a correct and precise pathological staging.

In consideration of our long-standing experience with PCND and the adoption of a standardized and reproducible technique [13], with this work, we report the incidence of CLNMs, the oncological outcomes and the complication rate in a population of patients treated for PTC and PCND. In particular, we focus on the evolution of the indications during the last decade, with the increased adoption of TL, in order to contribute to the understanding of the contemporary role of the PCND.

## 2. Methods

This is a retrospective single-center cohort study approved by the European Institute of Oncology Ethics Committee (IEO CODE: IEO2044). We selected a population of patients affected by PTC and treated by surgery, in a period going from 1998 to 2021. All patients underwent central neck dissection (CND), either prophylactic or therapeutic, unilateral or bilateral according to the clinical tumor stage.

Patients with the following characteristics were excluded from our analysis: age lower than 18 years, benign disease, previous thyroid treatment, histological type other than PTC, pT4 stage, presence of lateral neck metastases or distant metastasis at diagnosis, and follow-up <18 months.

Through the years, an ultrasound (US) study was performed by a senior radiologist in all patients who were candidates for treatment of thyroid cancer at our institute. According to the anatomically standardized IEO surgical procedure [13], a prophylactic central neck dissection was routinely performed on cN0 patients if the fine-needle aspiration cytology (FNAC) on the thyroid nodule was TIR3b/TIR4/TIR5 according to the Italian cytologic classification [14], corresponding, respectively, to the Bethesda IV, V and VI diagnostic categories [15]. The surgery consisted of TT or TL according to the tumor size and extension, number of nodules, US central lymph nodal status and patients’ characteristics. These indications changed according to the cumulating evidence of the past years (Table 1). Since 2015, after an internal audit, we have adopted TL and ipsilateral PCND for low-risk tumors (i.e., unilateral unifocal cT1-T2 with no evidence of extrathyroid extension and/or of neck node metastases at the preoperative institutional US). In any other cases, TT was the first choice, along with unilateral or bilateral prophylactic or therapeutic central neck dissection. Moreover, in 2018, we started to use thermal ablation or active surveillance in case of microcarcinoma [16], with the exception of nodules abutting the thyroid capsule or the region of the inferior laryngeal nerve or the esophagus.

The final decision about the extent of surgery was determined by a decision-making discussion in a multidisciplinary team and later shared with the patient.

After surgery, patients belonging to intermediate- and high-risk ATA group underwent I131 according to current guidelines [3,5,17]. In case of TL, the presence of central neck macrometastases or >5 micromestases and evidence of a T3 disease represent the contemporary indication for completion thyroidectomy and subsequent radioactive iodine ablation (RAI).

The following patients’ demographics, surgical and histopathologic details were retrieved from a thyroid dedicated IEO electronic database: age, sex, extension of surgery (TT vs. TL; unilateral vs. bilateral CND), histological result and pathological staging revised according to the 8th TNM staging [18], number and characteristics of CLNMs. Multifocality lesions were subdivided into three groups according to previous work [19]: multifocality confined to one thyroid lobe (group 1); unifocal lesion within each lobe (group 2); multifocality within both two lobes (group 3). The rate of post-surgical complications was retrieved as well (transient and permanent hypocalcemia; transient and permanent vocal cord paralysis). Permanent hypocalcemia was defined as the requirement of vitamin D and/or calcium supplementation for more than 6 months after surgery, established through parathormone and calcemia measurement. Permanent vocal cord paralysis was defined as a persistent paralysis at laryngeal endoscopic evaluation after 6 months.

A sub-analysis of selected clinical characteristics was performed upon patients with T1b and T2 carcinoma treated with TL with unilateral PCND.

Post-treatment surveillance includes physical examination, neck US, non-stimulated serum thyroglobulin, antithyroglobulin antibody (TgAb) and thyroid-stimulating hormone (TSH) concentration every 6 months. I^131^ scan or whole-body scan tomography were used for patients treated with I^131^ after 12 months.

### Statistical Analysis

Categorical variables were described as absolute and relative frequencies, while continuous variables were summarized as a median and interquartile range (IQR).

Categorical variables were compared using the Chi-square test or Fisher’s exact test when appropriate.

All survival outcomes were calculated starting from the date of surgery.

We considered the following survival endpoint:-Disease-free survival (DFS) includes all relapses of the thyroid disease and death from all causes.-Loco-regional recurrence-free survival (LRRFS) includes local and/or regional relapse of the thyroid disease and death from all causes.

LRRFS was defined as a new lesion in the thyroidectomy bed or cervical LNs detected through clinical examination, neck US, increased thyroglobulin levels and confirmed using FNAC or histopathological samples from the re-intervention. Metastases were detected using I^131^ scans, computed tomography or positron emission tomography (PET).

Survival was estimated using the Kaplan–Meier method. The impact on survival of different clinical and pathological factors (sex, age, pathological (p)T, macroscopic extracapsular extension, multifocality, pN, extranodal extension) was investigated with univariable and multivariable Cox proportional hazard models. A *p*-value < 0.05 was used as the threshold for statistical significance.

All statistical analyses were performed using R commander version 4.2.2, that is, a graphical user interface of R (The R Foundation for Statistical Computing, Vienna, Austria).

## 3. Results

From a database of 2890 patients, a total of 657 patients fulfilled the inclusion criteria. Descriptive results are summarized in Table 2. Briefly, the male–female ratio was 1:4 with a median age of 46.8 years. In most of cases, we performed a total thyroidectomy (66.8%). The central neck dissection was bilateral in 35.6% of cases. The majority of the population had a pT1a/b disease (80%). One hundred and thirty-one originally pT3 patients were re-staged as pT1 or pT2 according to the 8th TNM classification. Fifty-two patients who presented with clinical N1 (7.9%) and 13/52 (25%) were found to be negative (false positive). Pathological nodes were diagnosed in 240 of the 657 patients treated (36.5% of the population). In total, 41/240 patients (17.1%) had extranodal extension (ENE) and 64/240 had micrometastasis (26.6%). Ultimately, occult central node metastases were demonstrated in 201 patients of 657 (30.6%).

Age lower than 55 years, microscopic or macroscopic ETE and multifocality were found to be independent risk factors for CLNMs (Table 3).

I^131^ was administered to 307 patients (46.7%). The presence of CLNMs was the unique reason for an adjuvant treatment in 99 patients (15% of the total population). Of these, 81 patients (12.3%) were clinically N0, which means that they were upstaged after PCND. The presence of occult metastases in association with other pathological features (pT3, ETE, multifocality) represented the indication for I^131^ in 77 other patients (11.7%). 

Median follow-up was 78 months (48-114 months). At the end of follow-up, of the 657 patients analyzed, 16 died of other causes and 1 of disease. In addition, 5-year and 10-year DFS were 96.5% (94.7–97.7) and 93.3% (90.3–95.5), respectively. We did not find a difference between DFS and LRRFS in terms of events. In multivariate analysis, age (>55 years), pathological staging (pT) and extranodal extension (ENE) were independent factors associated with a worse DFS (Table 4). The rate of recurrence in the whole population was 2.7% (18 patients): 4 in the contralateral lobe, 2 in the central neck compartment, 10 in the lateral neck, and 2 in the lungs. One patient died of disease due to lung metastasis 5 years after the first surgery.

The rate of temporary and permanent vocal fold palsy was 12.8% and 1.8%, respectively, and it did not depend on the type of surgical procedure performed. Hypoparathyroidism was temporary in 42.2% and permanent in 11.9% of the patients, and it was significantly associated with TT compared with TL (*p* < 0.001) and bilateral CND compared with unilateral (*p* < 0.001) (Table 5). At least one parathyroid was found in 334 pathological specimens (51%) in the whole population.

Considering only patients classified as pT1b or pT2 according to the 8th TNM staging, 115 were treated with TL and unilateral PCND. Occult neck metastases were revealed in 30/115 patients (26.1%). Twenty-one had macrometastases (18.3%) and underwent completion thyroidectomy and I^131^. In this population, only three suffered of persistent hypoparathyroidism (2.6%).

## 4. Discussion

The treatment of well-differentiated thyroid carcinoma has been shifting towards more conservative approaches in the last decade [1,20], including active surveillance protocols [17,21,22] and image-guided thermal ablations [23] for very-low-risk cancer (unifocal intrathyroidal micropapillary carcinoma), while thyroid lobectomy has become the standard of care for clinical T1-T2 cancer [3,4,24,25].

With regard to central neck dissection for PTC, the role of a prophylactic procedure is still unproven and international guidelines are inconclusive, allowing each department to choose their strategy [5,17]. The 2015 ATA guidelines recommend a PCND in case of advanced tumors, clinically positive lateral neck nodes (cN1b), or if the lymph nodal status of the central compartment could be used for further treatment [3].

One of the main reasons for the lack of evidence is that studies comparing TT with PCND and TT alone have not had the statistical power to detect a difference in locoregional recurrence [26].

The arguments for and against PCND are known. Possible pros are as follows: better pathological staging for adjuvant treatment selection, association with lower post-surgical thyroglobulin (Tg) levels and effect on recurrence-free survival and disease-specific survival. The cons are as follows: unnecessary upstaging of some tumors, undefined prognostic role, and reported greater morbidity in terms of transient hypoparathyroidism [8].

In 2001, as a result of an internal audit about patients with nodal disease recurrence after thyroidectomy, we decided to incorporate CND as a routine procedure in the treatment of PTC, mainly on account of its potential predictive value for disease recurrence and a better staging and optimization of post-operative treatment and follow-up [27]. Later, we standardized our surgical technique and proposed a new anatomic classification of the neck level VI [13], subdivided into A-B-C-D areas, demonstrating the feasibility and safeness of a prophylactic procedure.

In this paper, we report the two-decade-long surgical experience with the adoption of PCND in a standardized fashion. In a homogeneous population treated for PTC pT1-T3, we found that about 30% of patients harbor occult metastases. This CLNM rate is unsurprising and even lower compared to other published studies, reporting up to 82.4% of cases of occult metastases [6]. It also confirmed two previous experiences reported by our group [27,28]. As demonstrated by numerous papers, age < 55 [29,30], pT [31,32], extracapsular extension [33,34,35] and multifocality have been shown to represent independent risk factors for central node metastases in PTC. Interestingly, even differentiating between various kind of multifocal presentation, we were not able to demonstrate a different metastasis rate in the three different kinds of multifocal disease, contrary to that found in the work of Heng et al. These authors observed a clinical impact in case of multifocality in at least one lobe, while unifocality in the two lobes did not correlate with CLNMs [19]. Despite a trend towards a major CLNM prevalence (40% vs. 29%), male sex did not impact significantly upon the CLNM rate, as described in other works [30,36,37]. Some of these risk factors have been grouped in nomograms with the purpose of selecting patients to be treated with PCND [30], incorporating, from time to time, other factors such as BRAF mutations [38] or the presence of US nodular calcifications or TgAb [39]. Nevertheless, most of these factors are available only after surgery and, for this reason, different groups are experimenting US nomograms models for better patient stratification with encouraging results [40].

We report a 96.5% 5-year DFS (94.7–97.7). With a median follow-up of 78 months, only 18 of 657 patients presented disease recurrence (2.7%). Of these, only two patients experienced a central neck compartment relapse, which is an extraordinarily low rate (0.3%) and confirms the high value of locoregional control of the PCND, which avoids re-operations and their associated higher rate of complications [41].

Considering our routine use of PCND, our result cannot be compared to a control group. One of the few prospective clinical trials presented in the literature comparing two groups (with or without PCND) did not find statistical difference in LRRFS in 181 patients (8% vs. 7.5%) [42]. Nevertheless, the group treated with only a thyroidectomy required a higher number of I^131^ courses (27.4% vs. 3.4%, *p* = 0.002) because of the evidence of elevated basal or stimulated Tg or for the presence of structural disease.

Another retrospective study comparing two homogeneous low-risk groups treated with and without PCND, selected through a score-matching system, did not find a significant difference in terms of local control. The reported rate of recurrence in these selected groups was too low to find any discrepancy (structural recurrence rate in two cases in the PCND vs. four cases 7.4% in the non-PCND) [43]. A different meta-analysis upon retrospective observational studies showed a significantly reduced locoregional recurrence rate for the TT with the PCND group compared with the TT group [6,44,45], while the only meta-analysis upon five randomized trial studies including 763 patients, predominantly with low-risk PTC, did not demonstrate a statistically significant effect of PCND on locoregional and biochemical recurrence [46]. Clearly, in a setting of low-level and conflicting evidence, answers could depend on the data interpretation and the weight given to the risks and benefits [47], and only ongoing clinical trials would bring stronger evidence in this debate [48].

Focusing on surgical complications, we report an overall 1.8% of permanent vocal cord palsy rate, which is in line with the published literature [6] and confirms the safety of this procedure for recurrent nerve integrity. On the other hand, we observed an overall 11.9% permanent hypocalcemia rate. This is a not-neglectable percentage, different from other published works, that prompts a few considerations. First, the definition of permanent hypocalcemia varies between studies in the literature. In our analysis, we included both groups of patients requiring either therapy with vitamin D alone or with calcium. Of 78 patients affected by post-operative hypoparathyroidism, 46 (59%) required a vitamin D integration only. In the literature, the permanent hypoparathyroidism rate varies between groups submitted to PCND: Hartl in 2012 reported a 0.6% rate in 317 patients, but the criteria for hypoparathyroidism definition were not known [37]. Popadich reported a 9.7% rate, adopting our definition [49]. In the work by Viola, persistent hypoparathyroidism was observed in 8% of patients in the group treated with TT alone compared to 19.4% in the group treated with PCND [42]. Despite a careful anatomical procedure, it is known that more extensive dissection in the central neck may interfere with the blood supply to the parathyroid glands, particularly the inferior ones [50]. Moreover, we found at least one parathyroid gland in more than half of the pathological specimens, highlighting a possible discerning difficulty between parathyroid and adipose tissue during the dissection itself. We tried to identify a possible trend in complications during the study period, but, in the end, we did not find any difference.

Through the refinement of surgical indications according to the update guidelines, we now reserve a conservative approach for cT1a diseases, adopting thermal ablations for single nodules without capsule involvement, waiting for new studies that will show the feasibility of this technique even for nodules larger than 1 cm or with capsule microinvasion. In this particular subgroup (T1a), we experienced only 3 of 123 relapses, all in the contralateral lobe. TL with unilateral PCND is employed in case of cT1b-T2 cN0 carcinoma. Hence, compared to other international experiences, we now differ only in this selected subgroup of patients, who take advantage of a unilateral resection. In our institute, cT3 lesions are always treated with TT and PCND, as indicated in most of the guidelines [3]. In the low-risk group (cT1b-T2), we observed a 26% rate of occult metastases. Interestingly, only 2.6% of these patients suffered from a persistent hypoparathyroidism.

Twelve percent of our patients were re-classified according to central nodal pathological stage after PCND and underwent iodine therapy. I^131^ was administered to 77 other patients (11.7%) for the presence of occult metastases in association with other pathological features (>pT2, ETE, multifocality). It is known that patients undergoing PCND are more prone to be selected for iodine therapy, as shown by Zhao in his meta-analysis (74.6% in the TT with PCND group vs. 59.9% in the TT group, OR 1.20, 95% CI 1.04–1.39) [6]. Despite a trend towards a reduction in the use of I^131^, the current guidelines still recommend adjuvant treatment for intermediate- and high-risk patients [3,5]. The presence of one macroscopic central node or more than five micrometastasic nodes (<2mm) leaves patients at intermediate risk of recurrence [3] and this is a reason for offering RAI regardless of age and stage [4,51]. Different papers deal with the risk of unnecessary upstaging and excessive use of iodine therapy, which could be associated with potential drawbacks from radiation, such as recurrent sialoadenitis, salivary gland swelling and increased risk of second primary malignancies in the long term [52]. Moreover, ongoing studies are trying to assess the real usefulness of adjuvant treatment in case of central neck pathological nodes [53]. Nevertheless, based on the current scientific knowledge, definitions such as “unnecessary upstaging” and “subclinical lymph nodes” attributed to occult node metastasis in PTC are questionable: leaving out the group of micrometastases, a lymph node is pathological independently on the means to detect it (US vs. PCND), and we have no justification for a different consideration, especially if we take into account the low sensitivity of US. For this reason, we can deem PCND not only a therapeutic procedure, but also a diagnostic and prognostic tool while waiting for future stronger evidence and updated guidelines. This is particularly interesting in case of diseases that are currently treated with TL: in our experience, almost 20% of patients in the cT1b-T2 group were upstaged after PCND, and they were selected for surgical totalization and adjuvant treatment.

We cannot ignore some limitations of our study, including the monocentric nature and retrospective analysis with potential selection, observational and data collection biases. Another limitation is the relatively small sample size and the absence of a control group. On the other hand, we report a long-standing standardized experience in a homogeneous group of patients and a helpful insight through the evolution of the surgical procedure.

## 5. Conclusions

Our results confirm the excellent results of adopting a routine use of PCND in terms of local control of the disease. Persistent hypoparathyroidism is a possible drawback of PCND, but this limit can be overcome through a further refining of patient selection. PCND is a safe procedure in case of T1b-T2 unilateral tumors and allows for a proper staging and selection of patients who necessitate adjuvant treatment.

## Figures and Tables

**Table 1 biomedicines-12-02177-t001:** Current treatment policy adopted at the Istituto Europeo Oncologico.

Staging	Policy
cT1a, cN0	Active surveillance Thermal ablation Thyroid lobectomy + ipsilateral central neck dissection
cT1b/cT2, cN0	Thyroid lobectomy + ipsilateral central neck dissection
cT1a/cT1b/cT2, cN1a	Total thyroidectomy + ipsilateral central neck dissection
cT3, cN0/cN1a	Total thyroidectomy + ipsilateral central neck dissection

**Table 2 biomedicines-12-02177-t002:** Clinical characteristics of the study cohort (657 patients).

Sex (%)	F	521 (79.3)
	M	136 (20.7)
Age (median, IQR)		46.8 [38.3, 55.8]
cN+ (central compartment) (%)	No	605 (92.1)
	Yes	52 (7.9)
Type of surgery (%)	Thyroid lobectomy	218 (33.2)
	Total thyroidectomy	439 (66.8)
pT 8th Ed AJCC (%)	1a	279 (42.5)
	1b	247 (37.6)
	2	93 (14.2)
	3a	23 (3.5)
	3b	15 (2.3)
Microcarcinoma (%)	No	446 (67.9)
	Yes	211 (32.1)
Central neck dissection (%)	Monolateral	423 (64.4)
	Bilateral	234 (35.6)
Multifocality (%)	No	461 (70.2)
	Yes	196 (29.8)
Type of multifocality (%)	Multi one lobe	88 (44.9)
	Single each lobe	52 (26.5)
	Multi each lobe	56 (28.6)
pN (%)	N0	417 (63.5)
	N1a	240 (36.5)
Extranodal extension (% on pN+)	No	199 (82.9)
	Yes	41 (17.1)
Total nodes in the dissection (median, IQR)		6.00 [3.00, 9.00]
Node micromestasis	Yes	64 (26.6)
Complication (%)	No	371 (56.5)
	Yes	286 (43.5)
Recurrent nerve section (%)	No	656 (99.8)
	Yes	1 (0.2)
Temporary vocal fold palsy (%)	No	573 (87.2)
	Yes	84 (12.8)
Permanent vocal fold palsy (%)	No	645 98.2)
	Yes	12 (1.8)
Temporary hypocalcemia (%)	No	380 (57.8)
	Yes	277 (42.2)
Permanent hypocalcemia (%)	No	576 (88.1)
	Yes	78 (11.9)
Completion thyroidectomy (%)	No	603 (91.8)
	Yes	54 (8.2)
Radioactive iodine (RAI) therapy (%)	No	350 (53.3)
	Yes	307 (46.7)
RAI for pN+ (%)	No	465 (70.7)
	Only for N	99 (15.1)
	N + other factors	93 (14.2)
Recurrence (%)	No	639 (97.3)
	Yes	18 (2.7)
State of disease (%)	NED	636 (96.8)
	AWD	4 (0.6)
	DOC	16 (2.4)
	DOD	1 (0.0)
Follow-up, months (median, IQR)		75 [48, 114]

Legend: AJCC American Joint Committee on Cancer, IQR interquartile range.

**Table 3 biomedicines-12-02177-t003:** Association between pN and clinical/pathological variables.

		pN0 (417)	pN1 (240)	*p*-Value
Sex (%)	F	336 (80.8)	185 (77.1)	0.317
	M	81 (19.2)	55 (22.9)	
Age categorized (%)	≤55 years	284 (68.3)	194 (80.8)	**0.001**
	>55 years	132 (31.7)	46 (19.2)	
Macroscopic extracapsular extension (%)	No	415 (99.5)	227 (94.6)	**<0.001**
	Yes	2 (0.5)	13 (5.4)	
Microscopic extracapsular extension (%) *	No	360 (86.7)	181 (79.7)	**0.020**
	Yes	55 (13.3)	46 (20.3)	
Multifocality (%)	No	308 (73.9)	153 (63.7)	**0.008**
	Yes	109 (26.1)	87 (36.2)	
Type of multifocality (%)	Multi one lobe	48 (44.0)	40 (46.0)	0.810
	Single each lobe	31 (28.4)	21 (24.1)	
	Multi each lobe	30 (27.5)	26 (29.9)	

* This analysis was conducted on the subgroup of patients without macroscopic extracapsular extension (642). Bold indicates results with significant impact.

**Table 4 biomedicines-12-02177-t004:** Cox proportional hazard models (univariable and multivariable).

**Univariable Models**			
**Variables**	**DFS**		
	**HR**	**CI 95%**	***p*-Value**
Sex			
▪ F	1		
▪ M	1.42	0.66–3.04	0.369
Age categorized			
▪ ≤55 years	1		
▪ >55 years	2.93	1.49–5.75	**0.002**
pT			**<0.001**
▪ T1	1		
▪ T2	1.65	0.66–4.11	0.284
▪ T3a	10.10	4.01–25.43	**<0.001**
▪ T3b	2.78	0.64–12.01	0.172
Multifocality			
▪ No	1		
▪ Yes	0.67	0.30–1.48	0.320
Macroscopic extracapsular extension			
▪ No	1		
▪ Yes	2.93	0.70–12.27	0.141
pN			
▪ N0	1		
▪ N1a	1.10	0.55–2.19	0.783
Extranodal extension			
▪ No	1		
▪ Yes	3.24	1.33–7.91	**0.010**
**Multivariable Models**			
**Variables**	**DFS**		
	**HR**	**CI 95%**	***p*-Value**
Age categorized			
▪ ≤55 y	1		
▪ >55 y	3.06	1.51–6.20	**0.002**
pT			<0.001
▪ T1	1		
▪ T2	1.58	0.63–3.94	0.329
▪ T3a	7.14	2.76–18.48	**<0.001**
▪ T3b	2.08	0.48–9.42	0.343
Extranodal extension			
▪ No	1		
▪ Yes	3.10	1.18–8.14	**0.021**

Legend: DFS = disease-free survival, HR = hazard ratio, CI = confidence interval. Bold indicates results with significant impact.

**Table 5 biomedicines-12-02177-t005:** Association between type of surgery and complications.

		**Thyroid Lobectomy** **(218)**	**Total Thyroidectomy** **(439)**	***p*-Value**
Temporary hypocalcemia	No	213 (97.7)	167 (38)	**<0.001**
	Yes	5 (2.3)	272 (62)	
Permanent hypocalcemia	No	214 (98.2)	362 (82.5)	**<0.001**
	Yes	4 (1.8)	74 (17.5)	
Temporary vocal fold palsy	No	202 (92.7)	371 (84.5)	**0.003**
	Yes	16 (7.3)	68 (15.5)	
Permanent vocal fold palsy	No	217 (99.5)	428 (97.5)	0.117
	Yes	1 (0.5)	11 (2.5)	
		**Unilateral CND** **(423)**	**Bilateral CND** **(234)**	***p*-Value**
Temporary hypocalcemia	No	305 (72.1)	75 (32.1)	**<0.001**
	Yes	118 (27.9)	159 (67.9)	
Permanent hypocalcemia	No	388 (91.8)	188 (80.3)	**<0.001**
	Yes	33 (8.2)	45 (19.7)	
Temporary vocal fold palsy	No	373 (88.2)	200 (85.5)	0.331
	Yes	50 (11.8)	34 (14.5)	
Permanent vocal fold palsy	No	417 (98.6)	228 (97.4)	0.363
	Yes	6 (1.4)	6 (2.6)	

Bold indicates results with significant impact.

## Data Availability

The data presented in this study are available on request from the corresponding author due to privacy and ethical reasons.

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
