# Peer review of "Central Neck Dissection in Papillary Thyroid Carcinoma: Benefits and Doubts in the Era of Thyroid Lobectomy"

_biomedicines, 2024, doi:10.3390/biomedicines12102177_

Round 1

Reviewer 1 Report

Comments and Suggestions for Authors

This manuscript reported the two-decade-long surgical experience with the adoption of PCND in a standardized fashion. The authors aimed to aimed to report our long[1]standing experience in PCND and its potential contemporary role in the treatment of PTC. In a homogeneous population treated for PTC pT1-T3, they found that about 30% of patients harbor occult metastases. PCND still represents a useful procedure that allows a very high local control rate and proper selection of patients needing adjuvant treatment, in particular those treated with a unilateral procedure. Bilateral approach is burdened by a not neglectable rate of permanent hypoparathyroidism.

1.       This manuscript used rigorous inclusion and exclusion criteria, and discussed the latest treatment recommendations thoroughly.

2.       From the 2020s, heat ablation also has been accepted as an option for PTCs with capsule micro-invasion, or larger than 1cm (T1b). Please discuss more.

3.       “The whole surgical experience (1998-2021) was divided in three different periods to outline a possible complication rate trend”. Could you provide the detailed grouping criteria and the basis for grouping, for instance, different guidelines or consensus?

4.       “Possible pros are: greater morbidity in terms of transient hypoparathyroidism.” I totally agree with this. In this manuscript, the mean follow-up was 78 months (48-114 months), only two patients experienced a central neck compartment relapse, which confirmed the high value in locoregional control of the PCND. But for T1a PTCs, local heat ablation is being Increasingly widely applied, we wonder the exact local recurrence rate in this group. Have you conducted any comparisons related to this problem?

Author Response

Cooment 1: This manuscript used rigorous inclusion and exclusion criteria, and discussed the latest treatment recommendations thoroughly.

Comment 2: From the 2020s, heat ablation also has been accepted as an option for PTCs with capsule micro-invasion, or larger than 1cm (T1b). Please discuss more.

Answer 2: This is a good point and we are waiting for more solid results from the literature in order to widen our indications for thermal ablation. We add a spot in the discussion section (page 6). Unfortunately we cannot discuss more about this topic due to article length reason.

Comment 3: “The whole surgical experience (1998-2021) was divided in three different periods to outline a possible complication rate trend”. Could you provide the detailed grouping criteria and the basis for grouping, for instance, different guidelines or consensus?

Answer 3: This analysis was made in order to outline a possible complication rate trend, according to the growing experience in this field and the employment of young surgeons. As it did not demonstrate any significant result and due to the arbitrary subdivision, we decided to exclude this analysis from the study.

Comment 4: “Possible pros are: greater morbidity in terms of transient hypoparathyroidism.” I totally agree with this. In this manuscript, the mean follow-up was 78 months (48-114 months), only two patients experienced a central neck compartment relapse, which confirmed the high value in locoregional control of the PCND. But for T1a PTCs, local heat ablation is being Increasingly widely applied, we wonder the exact local recurrence rate in this group. Have you conducted any comparisons related to this problem?

Answer 4:In table 4 we report our recurrence rate depending on T stage, showing that T3a lesion has a greater propensity to recur compared to other lesions. Generally, we had only three recurrences in T1a disease, all in the contralateral thyroid lobe, showing that this kind of disease is frequently indolent and can be controlled by non-invasive management. Unluckily we don’t have a comparison group of patients treated with thermal ablation in order to make some statement about the efficacy of the two different treatments. We added a spot in the discussion section (page 6).

Reviewer 2 Report

Comments and Suggestions for Authors

Dear Authors,

Your manuscript “Central neck dissection in papillary thyroid carcinoma: benefits and doubts in the era of thyroid lobectomy," biomedicines-3168927, presents your surgical experience with the adoption of PCND as an adjuvant treatment to classical TT or TL. The paper is written very well; it is easy to follow and contains some interesting observations and discussions. Therefore, I recommend its publication after one major and a few minor comments:

Major – Table 5 is missing. Please, include it

Minor:

1.        Abstract – once introduced, acronyms should be used afterwards (this goes for the term PTC). Additionally, the explanation of the terms ‘ETE’ and ‘DFS’ should be added.

2.        Abstract – the conclusion should be rewritten. Written in this way, it is too general and does not present your results properly. You should present shortly the conclusion that is based on your findings.

3.        The following lines should be cut out of the Statistical analysis section, and pasted above it, into Methods section:

“All survival outcomes were calculated starting from the date of surgery.

We considered the following survival endpoints.

- Disease Free Survival (DFS) includes all relapses of the thyroid disease and death from all causes.

- Loco-Regional Recurrence Free Survival (LRRFS) includes local and/or regional re-lapse of the thyroid disease and death from all causes.

LRRFS was defined as a new lesion in the thyroidectomy bed or cervical LNs detected through clinical examination, neck US, increased thyroglobulin levels and confirmed using FNAC or histopathological samples from the re-intervention. Metastases were detected using I131 scans, computed tomography or positron emission tomography (PET).”

4.        Please explain how you tested the independence of the factors. The statistical tests applied should be added to the Statistical analysis section.

5.        The legend of Table 2 contains the title of Table 3. Please delete it.

6.        Pg 4 – It is written: “Age less than 55 years, microscopic or macroscopic ETE and multifocality resulted as independent risk factors for CLNM (Table 3).” However, Table 3 presents the correlations between pN and independent variables. Please clarify what table 3 presents and correct the text properly. If the table presents correlations, then the correlation’s coefficients should be introduced to the table. If the table presents the independency of the factors tested, then appropriate methodology should be named in the Statistical analysis section and under the table.

7.        Pg 4 (the last line) - the explanation of the term ‘LRR’ should be added.

8.        Conclusion – The conclusion should be highlighted, presented in a separate section in a clearer way. You may utilize some lines from the discussion section (e.g. merge and paraphrase the 2nd paragraph and the last line on page 6).

Author Response

  1. Abstract – once introduced, acronyms should be used afterwards (this goes for the term PTC). Additionally, the explanation of the terms ‘ETE’ and ‘DFS’ should be added.

Anwer 1: Thanks for your comment, we corrected it.

  1. Abstract – the conclusion should be rewritten. Written in this way, it is too general and does not present your results properly. You should present shortly the conclusion that is based on your findings.

Answer 2: Thanks for your comment, we amended the conclusion section of the abstract in order to make it more consistent with our results.

  1. The following lines should be cut out of the Statistical analysis section, and pasted above it, into Methods section:

“All survival outcomes were calculated starting from the date of surgery.

We considered the following survival endpoints.

- Disease Free Survival (DFS) includes all relapses of the thyroid disease and death from all causes.

- Loco-Regional Recurrence Free Survival (LRRFS) includes local and/or regional re-lapse of the thyroid disease and death from all causes.

LRRFS was defined as a new lesion in the thyroidectomy bed or cervical LNs detected through clinical examination, neck US, increased thyroglobulin levels and confirmed using FNAC or histopathological samples from the re-intervention. Metastases were detected using I131 scans, computed tomography or positron emission tomography (PET).”

Answer 3: The statistical analysis is part of the methods section. In our opinion, it is clearer to group all the explanations of the methods that we adopted for the statistical analysis together with the oncologic outcomes definition in this paragraph.

  1. Please explain how you tested the independence of the factors. The statistical tests applied should be added to the Statistical analysis section.

Answer 4: We used Paerson’s Chi-square to test the independence of the factors. We added it in the statistical analysis section.

  1. The legend of Table 2 contains the title of Table 3. Please delete it.

Answer 5: done.

6. Pg 4 – It is written: “Age less than 55 years, microscopic or macroscopic ETE and multifocality resulted as independent risk factors for CLNM (Table 3).” However, Table 3 presents the correlations between pN and independent variables. Please clarify what table 3 presents and correct the text properly. If the table presents correlations, then the correlation’s coefficients should be introduced to the table. If the table presents the independency of the factors tested, then appropriate methodology should be named in the Statistical analysis section and under the table.

Answer 6: Thanks for your suggestion. We erroneously used the term “correlation” and “independent”. We modified the title.

  1. Pg 4 (the last line) - the explanation of the term ‘LRR’ should be added.

Ansewr 7: We corrected it.

  1. Conclusion – The conclusion should be highlighted, presented in a separate section in a clearer way. You may utilize some lines from the discussion section (e.g. merge and paraphrase the 2nd paragraph and the last line on page 6).

Answer 8: We followed your advice and created an indipendent section dedicated to conlusions making it clearer and shorter.

Round 2

Reviewer 2 Report

Comments and Suggestions for Authors

Dear Authors,

The corrected version of your manuscript “Central neck dissection in papillary thyroid carcinoma: benefits and doubts in the era of thyroid lobectomy," biomedicines-3168927, is significantly better; but it is insufficiently improved because you did not provide an answer or make any corrections, along with the major remark that Table 5 is absent. Please add Table 5 or remove the appropriate text from the Results section (page 4).

Aside from this fundamental point, there is another: the statistical analysis is not performed adequately.

The x2-test is a statistical test that can demonstrate the association between tested factors (which is corrected in an appropriate way in Table 3 in the revised version of your manuscript), but it cannot be used to test the independence of tested factors. Some multivariate tests should be applied for these purposes. Please utilize the right statistical test and correct accordingly the Material and Method section, along with the Results section and Table 3.

Author Response

1) The corrected version of your manuscript “Central neck dissection in papillary thyroid carcinoma: benefits and doubts in the era of thyroid lobectomy," biomedicines-3168927, is significantly better; but it is insufficiently improved because you did not provide an answer or make any corrections, along with the major remark that Table 5 is absent. Please add Table 5 or remove the appropriate text from the Results section (page 4).

Answer 1: Thank you for the warning, we added table 5 that was missing from the original text.

2) Aside from this fundamental point, there is another: the statistical analysis is not performed adequately.

The x2-test is a statistical test that can demonstrate the association between tested factors (which is corrected in an appropriate way in Table 3 in the revised version of your manuscript), but it cannot be used to test the independence of tested factors. Some multivariate tests should be applied for these purposes. Please utilize the right statistical test and correct accordingly the Material and Method section, along with the Results section and Table 3.

Answer 2: Thank you for the accurate comment. Indeed, our previous sentence was misleading, so we decided to rephrase it. We included in the survival analysis all the factors that could potentially be significant, from current literature and personal experience. We hope that the new sentence is clear and acceptable.

Round 3

Reviewer 2 Report

Comments and Suggestions for Authors

Dear Authors,

Version 3 of your manuscript, “Central neck dissection in papillary thyroid carcinoma: benefits and doubts in the era of thyroid lobectomy," biomedicines-3168927, is sufficiently improved; therefore, I recommend its publication after one minor correction:

Table 5 presents associations between type of surgery and complications, not correlations. Please add the statistical test applied under the table and correct accordingly the Material and Method section, along with the Results section and Table 5.

Author Response

Version 3 of your manuscript, “Central neck dissection in papillary thyroid carcinoma: benefits and doubts in the era of thyroid lobectomy," biomedicines-3168927, is sufficiently improved; therefore, I recommend its publication after one minor correction:

Table 5 presents associations between type of surgery and complications, not correlations. Please add the statistical test applied under the table and correct accordingly the Material and Method section, along with the Results section and Table 5.

Answer:

Thank you for your comment. We corrected the table 5 title and the results section accordingly. The test applied was Chi-square test and this is specified in the methods section-statystical analysis: "Categorical variables were compared using Chi-square test or Fisher’s exact test when appropriate". If you prefer that we specify the statystical test adopted under the table, we can add it for every table we are presenting in our paper.